# Nitric Oxide Pre-Treatment Advances Bulblet Dormancy Release by Mediating Metabolic Changes in *Lilium*

**DOI:** 10.3390/ijms26010156

**Published:** 2024-12-27

**Authors:** Chenglong Yang, Xiaoping Xu, Muhammad Moaaz Ali, Xing He, Wenjie Guo, Faxing Chen, Shaozhong Fang

**Affiliations:** 1Institute of Biotechnology, Fujian Academy of Agricultural Sciences, Fuzhou 350003, China; yangchenglong9004@163.com (C.Y.); xuxiaoping@faas.cn (X.X.); 18366197898@163.com (X.H.); guowenjie@faas.cn (W.G.); 2The School of Tropical Agriculture and Forestry, Hainan University, Danzhou 571700, China; muhammadmoaazali@yahoo.com; 3College of Horticulture, Fujian Agriculture and Forestry University, Fuzhou 350002, China

**Keywords:** *Lilium*, dormancy, nitric oxide, gibberellin, abscisic acid, sodium nitroprusside

## Abstract

The lily is a globally popular cut flower, and managing dormancy in lily bulblets is essential for continuous, year-round production. While nitric oxide (NO) has been shown to influence seed dormancy and germination, its role in dormancy release in lilies was previously unconfirmed. In this study, we investigated the effects of NO on dormancy release in lily bulblets using SNP and c-PTIO. Results showed that SNP treatment promoted dormancy release, while c-PTIO inhibited it. Measurement of endogenous NO levels in the bulbs, along with enzyme activities of NOS-like and NR and gene expression levels of *LoNOS-IP* and *LoNR*, confirmed that NO plays a role in promoting dormancy release in lilies. To further elucidate the physiological mechanisms involved, we analyzed H_2_O_2_ levels, antioxidant enzyme activities, endogenous hormone levels, and carbohydrate metabolism in the bulbs. Findings demonstrated that NO facilitated dormancy release by increasing H_2_O_2_, gibberellins (GAs), indole-3-acetic acid (IAA), zeatin riboside (ZR), reducing sugars, and by accelerating the metabolism of abscisic acid (ABA) and starch. This study provides a foundation for deeper investigation into the mechanisms underlying dormancy release in lily bulbs.

## 1. Introduction

Lilies (*Lilium* spp.) are widely cherished geophytes in the field of horticulture, serving as popular choices for cut flowers, potted plants, and garden adornments [1,2]. Dormancy, a crucial adaptive mechanism, plays a key role in the survival of lilies in adverse conditions [3,4]. Belonging to the family *Liliaceae*, the genus *Lilium* consists of over 100 species distributed across cold and temperate regions of the northern hemisphere [5]. Various populations and varieties exhibit distinct dormancy traits as they adapt to their respective habitats [6,7]. In their natural cycle, lilies sprout in spring and blossom in summer. Failure to meet the chilling requirements of lilies, particularly in regions with mild winters, can result in erratic budbreak and asynchronous growth, ultimately leading to reduced vegetative development and sporadic flowering [8,9].

Cold storage treatment is widely recognized as the most effective method for inducing dormancy release [10]. Consequently, the year-round demand for flowers has prompted farmers to manipulate flowering times by subjecting lily bulbs to cold storage post-harvest, effectively mimicking winter conditions [11]. However, the increased commercial costs associated with low-temperature treatment pose a barrier to commercial production, and the extensive use of cold storage facilities is anticipated to contribute to the issue of global warming. As a result, alternative methods for low-temperature treatment have emerged as a significant area of focus in lily research.

Dormancy is a complex physiological process, involving various morphological, physiological, biochemical, and transcriptional events [12,13]. Phytohormones play a crucial role in regulating dormancy release [14,15]. The antagonistic relationship between abscisic acid (ABA) and gibberellin (GA) is considered a key factor in dormancy. Cold treatment over a long period contributes to reducing ABA levels in dormant organs and increasing GA content, aiding in dormancy release. Numerous studies have demonstrated the significant positive impact of GA_3_ treatment on breaking dormancy in *Lilium* cultivars [1,16,17,18,19]. ABA is recognized as a vital phytohormone in dormancy regulation, particularly as a central hub in seed dormancy [20]. It is known to be involved in dormancy regulation by interpreting temperature signals, and external ABA treatment can enhance bulb dormancy and suppress bulb germination in lilies [21].

Energy metabolism also plays a role in dormancy regulation [22]. Lily bulbs primarily contain starch and glucomannan, which are broken down into monosaccharides after exposure to cold, providing fuel for the sprouting process [23]. The activities of enzymes like α- and β-amylase are linked to dormancy release, with low-temperature storage resulting in increased activities of various starch-degrading enzymes [24,25].

Reactive oxygen species (ROS), particularly hydrogen peroxide (H_2_O_2_), play a central role in the signal transduction pathways that enable plants to adapt to environmental changes. In conditions of oxidative eustress, a physiological steady-state of H_2_O_2_ is maintained at low, nanomolar levels. Optimal levels of ROS have been found to facilitate bulblet dormancy release. Conversely, elevated ROS levels in bulbs can trigger an increase in antioxidant enzyme activity, safeguarding cells from oxidative damage caused by oxygen radicals. Thus, appropriate ROS concentrations are crucial for promoting plant dormancy release [26,27]. During long-term cold treatment, key antioxidants like superoxide dismutase (SOD), catalase (CAT), and peroxidase (POD) exhibited varying activity changes, impacting the levels of H_2_O_2_ and ultimately accelerating dormancy release [28]. A recent study revealed that the *LoNFYA7–LoVIL1* module plays a crucial role in orchestrating the transition from slow to fast growth in lily bulbs, suggesting that *LoVIL1* can serve as a reliable marker for the bud-growth-transition trait post-dormancy release in lily cultivars [29]. Additionally, chemical treatments like 2,3-Butanedione oxime (BDM), N-Ethyl maleimide (NEM), and 2-Deoxy-D-glucose (DDG) have shown promise in regulating bulblet dormancy [30]. These findings highlight the strong correlations between physiological metabolism and dormancy release in lily bulbs during cold treatment.

Nitric oxide (NO) is known to play a crucial role in dormancy release and the seed germination process [31,32]. NO acts as a key gaseous molecule in promoting seed germination by regulating ABA metabolism and the GA synthesis pathway [33]. There are two pathways for NO synthesis in plants: one through a nitrate/nitrite-dependent pathway, and the other through an NO synthase (NOS)-mediated oxidation pathway [34]. Nitrate reductase (NR) is a multifunctional cytoplasmic enzyme that plays a critical role in nitrogen assimilation and metabolism. It catalyzes the rate-limiting step of nitrate assimilation by reducing nitrate to nitrite, using NADH as an electron donor. The production of NO by NR in plant physiology has been extensively demonstrated through both pharmacological and genetic approaches. In contrast, in animal systems, the majority of NO production is mediated by NOS, which catalyzes the oxygen- and NADPH-dependent oxidation of L-arginine to NO and citrulline [35,36].

Sodium nitroprusside (SNP) is an NO donor widely used to demonstrate the positive regulation of NO during dormancy release. Conversely, 2-(4-carboxphenyl)-4,4,5,5-tetramethylimidazoline-1-oxyl-3-oxide (c-PTIO) is used as an NO scavenger to inhibit dormancy release. In potato, a 40 uM SNP treatment promoted tuber sprouting, while c-PTIO repressed the influence of NO on tuber sprouting [37]. A 3 mM SNP treatment promoted Apple (*Malus domestica Borkh*.) embryo germination, whereas a 0.3 mM c-PTIO treatment significantly inhibited germination [38]. Exogenous SNP treatment also promoted seed germination in Indian mustard (*Brassica juncea* L.), *Arabidopsis*, and *Oryza sativa* [39,40,41]. Recent studies have shown that exogenous NO promotes dormancy release by interacting with phytohormone signaling pathways, such as GA catabolism, ABA catabolism, and affecting the accumulation of ROS and the synthesis of enzymes related to the antioxidant system [42,43,44]. While there have been extensive studies on the effects of NO on the physiological progress of dormancy release, the mechanisms of dormancy release in different plants vary. Whether NO regulates dormancy release in lily remains an open question.

Here, in this study, the lily bulblets responded to SNP treatment with accelerated dormancy release and earlier sprouting through the modulation of enzyme activities, NO and H_2_O_2_ levels, and changes in carbohydrate metabolism and phytohormone content. These physiological shifts were accompanied by altered expression of specific genes related to dormancy and growth regulation. Conversely, treatment with c-PTIO, an NO scavenger, inhibited these responses, reinforcing the role of nitric oxide in regulating dormancy release and promoting sprouting in lily bulblets. These findings indicate that NO treatment has the potential to partially substitute for low-temperature storage, offering practical applications in the cut-flower industry by reducing cold storage electricity usage, lowering enterprise costs, and minimizing greenhouse gas emissions.

## 2. Results

### 2.1. Phenotypic Analysis

SNP and c-PTIO are commonly used to assess the functional effects of NO on plant physiology. To determine if NO is involved in bulb dormancy release, harvested lily bulbs were treated with varying concentrations of SNP and c-PTIO. In prior studies, bud length of lily bulbs has been a standard indicator of dormancy release timing, where longer buds signify an earlier dormancy release [21,45]. As shown in Figure 1, SNP treatments at 5 mM, 10 mM, and 20 mM promoted bulb dormancy release. Compared to the control (CK), SNP-treated bulbs showed significant increases in bud length, with the 20 mM treatment group exhibiting the longest buds and the 5 mM group the shortest. Conversely, treatment with 1 mM c-PTIO significantly inhibited dormancy release. When applied alongside SNP, c-PTIO’s inhibitory effect was notably reduced. These findings suggest that SNP treatment facilitates bulb dormancy release, while c-PTIO suppresses it.

### 2.2. Effect of SNP and c-PTIO on Endogenous NO Content

To explore whether SNP and c-PTIO influence bulb dormancy release by modulating endogenous NO content, we measured NO content, enzyme activities, and the expression levels of related genes in samples under different treatments. Figure 2 illustrates a distinct trend in NO content for SNP-treated samples, showing an initial increase followed by a decline. NO content significantly rose within 4 h of SNP treatment, peaking at 8 h. In contrast, c-PTIO treatment significantly reduced the NO increase. Additionally, co-treatment with SNP and c-PTIO partially offset the rise in NO levels. SNP treatment also led to a marked increase in the activities of NOS-like enzymes and Nitrate Reductase (NR), whereas c-PTIO notably suppressed these activities. qRT-PCR analysis showed that expression levels of *LoNOS-IP* and *LoNR* genes were significantly upregulated after 4 h of SNP treatment compared to the control group. These results indicate that SNP and c-PTIO regulate dormancy release by influencing endogenous NO content.

### 2.3. Analysis of H_2_O_2_ and Antioxidant Activity

The interaction between NO and ROS formation is crucial during dormancy release. Studies have linked dormancy alleviation with increased H_2_O_2_ accumulation [46,47]. Throughout lily dormancy release, H_2_O_2_ levels in the SNP treatment group consistently exceeded those in other groups, indicating that SNP treatment elevates H_2_O_2_ in dormant buds. Conversely, c-PTIO, acting as an NO scavenger, reduced H_2_O_2_ levels. Figure 3 shows that the peak H_2_O_2_ content in dormant buds following SNP treatment occurred 10 days earlier than in other groups, correlating with the earlier dormancy release shown in Figure 1. These findings suggest that SNP’s dormancy release effect is linked to changes in H_2_O_2_ content within dormant buds.

Superoxide dismutase (SOD), a primary enzyme for scavenging superoxide anions (O_2_⁻), catalyzes their dismutation into H_2_O_2_. The pattern of SOD activity during dormancy release in Figure 3C aligns with H_2_O_2_ variations. During early dormancy, SNP significantly increased peroxidase (POD) activity in dormant buds, while c-PTIO reduced it relative to the control group. Figure 3D illustrates that while POD scavenges H_2_O_2_, H_2_O_2_ levels continued to rise, suggesting that POD is not the primary enzyme regulating H_2_O_2_ content. After SNP treatment, the sharp decline in H_2_O_2_ content coincided with reduced POD activity, implying that changes in POD activity are driven by H_2_O_2_ variations.

In early dormancy stages, SNP treatment also enhanced catalase (CAT) activity, while c-PTIO suppressed it (Figure 3A). However, by late dormancy, neither treatment significantly affected CAT activity, suggesting that NO facilitates H_2_O_2_ accumulation through mechanisms other than CAT inhibition. This suggests that NO may enhance H_2_O_2_ accumulation through alternative pathways.

### 2.4. Changes in Endogenous Hormones During Dormancy Release

In the early stages of dormancy, SNP treatment significantly reduced endogenous ABA content in the bulbs, with levels progressively decreasing throughout the dormancy period and reaching a minimum at 40 days post-treatment. While both the control and c-PTIO-treated groups exhibited a similar trend in ABA reduction, their lowest levels occurred later than those in the SNP group. As shown in Figure 4, SNP treatment markedly upregulated *LoCYP707A1* expression while downregulating *LoNCED1* expression. Conversely, SNP treatment significantly elevated endogenous GA content in early dormancy stages, with a rapid increase between 30 and 40 days, earlier than in other treatment groups. This was accompanied by increased *LoGA20ox* expression and suppressed *LoDELLA* gene expression.

In the case of endogenous IAA, SNP treatment initially reduced its content in early dormancy, but IAA levels rapidly rose after 20 days, surpassing other groups by day 40. The expression pattern of the auxin-related gene *LoGH3.1* aligned with the changes in IAA levels. Endogenous ZR levels across treatments followed a similar trend to endogenous GA, but there were no significant differences in ZR content among groups during early dormancy.

### 2.5. Influence of Carbohydrate Metabolism

During dormancy release, the starch content in lily buds gradually declined across all treatment groups, while reducing sugar content steadily increased. SNP treatment promoted an earlier increase in reducing sugars and accelerated the decrease in starch content, with the highest level of reducing sugars reached 40 days post-treatment—10 days earlier than in the control group. Concurrently, starch content in the SNP group hit its lowest point. These findings suggest that NO facilitates earlier dormancy release by accelerating starch breakdown and enhancing reducing sugar accumulation.

Throughout the experiment, the expression of genes *LoAMY*, *LoBMY*, *LoSPS*, *LoTMT*, and *LoSUS* generally increased in early dormancy, peaking before declining in later stages. Peak expression in the SNP-treated group occurred 10 days earlier than in the control, while in the c-PTIO group, the peak was delayed by 10 days. For the LoSUT gene, expression decreased initially in early dormancy and increased in later stages. The lowest expression value for *LoSUT* in the SNP-treated group appeared 10 days earlier than in the control, whereas in the c-PTIO group, it was delayed by 10 days (Figure 5).

## 3. Discussion

Nitric oxide (NO) has been shown to effectively promote dormancy release or germination in various plant species [48,49,50]. Previous studies suggest that the improvement in germination of dormant seeds through exogenous treatments, such as nitrogen compounds like nitrates and nitrites, is likely facilitated by NO production [51]. The role of NO in plant physiology has mainly been demonstrated through pharmacological experiments using NO donors and/or NO scavengers. Due to NO’s toxicity, reactivity, and gaseous nature, its direct application in the lab is challenging. Therefore, compounds like sodium nitroprusside (SNP) are commonly used to generate NO. However, the photolysis of SNP can release more cyanide than NO, indicating that cyanide might be the active compound when SNP is applied to seeds.

In the present study, to account for the potential side effects of SNP, we also employed c-PTIO in our experiments. The contrasting effects of NO donors and NO scavengers in a specific physiological process often provide strong evidence for NO’s involvement. SNP treatment promoted dormancy release by increasing radicle length in seed bulbs, whereas c-PTIO inhibited this effect, with no significant difference observed in the SNP + c-PTIO treatment group (Figure 1). However, dormancy release in bulbous plants differs from seed dormancy release and germination. To further explore the relationship between NO and dormancy release in lilies, we measured the endogenous NO content in the treated bulbs. SNP treatment significantly increased endogenous NO content, while c-PTIO had the opposite effect (Figure 2). These results provide compelling evidence for the regulatory role of NO in dormancy release.

It is widely acknowledged that ROS have emerged as pivotal regulators in the dormancy and germination of seeds. Depending on their concentrations, ROS can serve as positive signaling agents, including the promotion of dormancy release and germination, or they may lead to detrimental outcomes [52]. Under cold stress, plant cells rapidly produce certain reactive oxygen molecules, such as superoxide O_2_^−^ and H_2_O_2_. NO can react with O_2_^−^ to form the highly destructive peroxynitrite anion (ONOO^−^) [53]. Furthermore, NO can modulate the levels of H_2_O_2_ by affecting the activity of antioxidant enzymes. In this study, treatment with SNP enhanced the activities of SOD and POD, and promoted the endogenous content of H_2_O_2_ [54]. This result suggests that the key role of NO in dormancy release may be attributed to its crosstalk with reactive oxygen species.

It is well known that the phytohormone ABA often plays a key role in the induction of seed dormancy [55]. It has been reported that the interaction of NO and ABA in the regulation of seed germination has been demonstrated in plants such as Arabidopsis, switchgrass and warm-season C4 grasses [56,57]. NO acts upstream of ABA and GA biosynthesis, enhancing GA production while inhibiting ABA biosynthesis to promote seed germination. Recent studies have shown that in potato tubers, NO generated by NOS or NR triggers the expression of the ABA catabolic gene *StCYP707A1* and suppresses the ABA biosynthesis-related gene *StNCED1*, leading to decreased ABA levels and a shift in the ABA-GA balance that induces tuber sprouting [37]. Our study similarly found that NO upregulated the ABA catabolic gene *LoCYP707A1* (Figure 4F) and downregulated the ABA biosynthesis gene *LoNCED1* (Figure 4J), resulting in reduced endogenous ABA levels (Figure 4A).

Additionally, NO increased the expression of the GA biosynthesis gene *LoGA20ox* (Figure 4H) and decreased the expression of the GA signaling repressor gene *LoDELLA* (Figure 4I), leading to higher endogenous GA levels (Figure 4B), thus accelerating the dormancy release. Typically, auxins and cytokinin play a crucial role in the dormancy process of lily bulblets [58]. It is evident from this study that we observed a rapid increase in the endogenous levels of IAA (Figure 4C) and ZR (Figure 4D) promoted by NO, thereby accelerating the release from dormancy. The expression of the gene *LoVIL*, a marker for lily dormancy, is commonly used to assess the bulblet dormancy process [29]. The promotion of *LoVIL* expression by NO (Figure 4K) is consistent with the observed phenotype. However, the interaction between NO and plant hormones during the release from dormancy in lily bulblets is highly complex, and the underlying mechanisms warrant further investigation.

Changes in carbohydrate metabolism have been reported to be associated with dormancy release during the cold storage of lily bulbs. Bulbs exposed to low temperatures induced a net breakdown of starch and the accumulation of soluble sugars in the bulb scales [3]. In the present study, the starch content in the bulblets gradually decreased during low-temperature storage (Figure 5A), while the content of reducing sugars gradually increased (Figure 5B). Interestingly, treatment with NO accelerated the degradation of starch and the increase in reducing sugars. Furthermore, the expression levels of the *LoAMY* and *LoBMY* genes significantly increased after NO treatment (Figure 5C,D). The *LoSUT* gene, responsible for the long-distance transport of major carbohydrates, has been shown to regulate the dormancy and vernalization processes in lilies [59]. The *LoTMT* gene, acting upstream of *LoSUT* and involved in the loading and unloading of sugars in the phloem, was also observed to have its expression promoted by NO in this study, with *LoTMT* exhibiting an expression pattern that precedes *LoSUT* by one cycle. These results suggest that NO treatment can regulate carbohydrate metabolism in lily bulblets, thereby accelerating the process of dormancy release (Figure 6). However, how sugars participate in the perception process and signal transduction remains largely unknown, and the specific regulatory mechanisms of NO on carbohydrate metabolism warrant further investigation in subsequent studies.

## 4. Materials and Methods

### 4.1. Plant Material and Applied Treatment

The oriental hybrid lily cultivar ‘Siberia’ was used as the plant material in this study. Lily bulblets were harvested from a greenhouse at the Institute of Biotechnology, Fujian Academy of Science, Fuzhou, China. Only bulblets that were externally undamaged, uniform in size, and with a circumference of 10–12 cm were selected. These bulblets were disinfected by soaking in a 500-fold dilution of 50% thiophanate-methyl wettable powder for 30 min, then placed in a shaded area to air-dry for experimental use.

The pretreatment experiment with SNP and c-PTIO followed the method described by Gniazdowska et al. [45]. The appropriate concentrations of SNP and c-PTIO were selected based on a preliminary experiment, with the details provided in Appendix A. Bulblets were placed into glass beakers containing SNP solutions at concentrations of 5 mM, 10 mM, and 20 mM, as well as a 1 mM solution of c-PTIO. Control samples were treated with distilled water instead. After 3 h of treatment in darkness at 25 °C, the beakers were opened to release any gaseous products. The bulbs were then rinsed 3–4 times with distilled water and transferred to plastic baskets filled with peat soil. They were wrapped in plastic film and stored at 4 °C in cold storage.

### 4.2. Germination Tests

Previous studies [21,60] have shown that smaller bulblets of Oriental lilies can effectively exit dormancy and produce visible buds after 60 days of cold storage. Following this period, 30 lily bulblets were randomly selected from each SNP treatment group, the c-PTIO treatment group, and the control group for statistical analysis. Dormancy release timing was determined by measuring the length of the lily buds (from the basal plate to the tip of the visible buds); longer sprouts indicated an earlier dormancy release among the different treatment groups.

### 4.3. Measurement of Enzyme Activities

The method for measuring NR activity followed Lu et al. [61]. Approximately 1.0 g of buds was ground in a mortar, and 10 mL of 50 mM phosphate buffer (pH 7.8) was added. The homogenate was then centrifuged at 12,000 rpm for 20 min, and the supernatant was transferred to a fresh centrifuge tube. Nitrite content was determined by measuring absorbance at 540 nm using a UV–visible spectrophotometer (Shanghai Metash Instruments Co., Ltd., Shanghai, China). NOS-like activity was measured using a plant NOS ELISA kit (Beijing Haolesi Biotechnology Co., Ltd., Beijing, China) according to Diao et al. [62].

### 4.4. Determination of NO and H_2_O_2_ Contents

The NO content was measured using a plant nitric oxide enzyme-linked immunosorbent assay (ELISA) kit (Shanghai Hanhong Biotechnology Co., Ltd., Shanghai, China, FT-P5118Z) following the manufacturer’s instructions. H_2_O_2_ content determination followed the method from a previous study [63]. Buds were rinsed with deionized water, surface moisture was absorbed with filter paper, and the buds were weighed. Fresh buds (0.5–0.6 g) were homogenized in 5 mL of propanone. The mixture was adjusted to a final volume of 10 mL, allowed to stand for 5 min, and centrifuged at 5000 rpm for 10 min. The resulting supernatant was combined with 0.1 mL of 5% Ti(SO_4_)_2_ (titanium sulfate) and 0.2 mL of NH_3_ (ammonia), then centrifuged at 10,000 rpm for 10 min at 4 °C. The absorbance of the resulting solution was measured at 415 nm (OD_415_).

### 4.5. Phytohormones Measurements

Approximately 1 g of buds was collected at each sampling time, quickly frozen in liquid nitrogen, and stored at −80 °C. Endogenous hormone content was measured using plant hormone (GA, ABA, IAA, ZR) enzyme-linked immunosorbent assay (ELISA) kit (Shanghai Chaorui Biotechnology Co., Ltd., Shanghai, China, HP-E21782, HP-E21768, HP-E21773, HP-E21752) following the manufacturer’s instructions.

### 4.6. Carbohydrate Assay

Starch content was determined using the iodine colorimetric method [64]. Lily buds (0.5 g) were thoroughly ground in 2 mL of distilled water, followed by the addition of 3.2 mL of 60% perchloric acid, and ground for an additional 10 min. The mixture was then centrifuged at 5000× *g* for 5 min at room temperature, and the supernatant was filtered to obtain the starch extract. For analysis, 0.5 mL of the extract solution was combined with 3 mL of distilled water and 2 mL of iodine reagent, allowed to stand for 5 min, and then diluted to 10 mL with distilled water. Absorbance was measured at 660 nm.

Soluble sugar content was measured following the method described by Wei et al. [64]. Buds were ground into a fine powder and transferred into a test tube. After a 30-min hydrolysis in a boiling water bath, the mixture was filtered and diluted. A 0.5 mL aliquot of the extract solution was added to a tube containing 1.5 mL of distilled water, 0.5 mL of anthraquinone-ethyl acetate reagent, and 5 mL of concentrated H_2_SO_4_. The tubes were shaken in a boiling water bath for 1 min, and absorbance was measured at 630 nm.

### 4.7. RNA Extraction and Quantitative Reverse Transcriptase-Polymerase Chain Reaction

Total RNA from the middle of the underground stem node was isolated at different growth stages of lily using an RNAprep Pure Plant Kit (TianGen Biotech, Beijing, China), following the kit protocol. RNA degradation and contamination was monitored on 1% agarose gel. RNA quantification and purity assessment were performed spectroscopically using the NanoPhotometer^®^ spectrophotometer (NanoDrop Technologies, Wilmington, DE, USA). cDNA synthesis was performed using a ReverAid First Strand cDNA Synthesis Kit (Thermo Fisher, Waltham, MA, USA) according to the manufacturer’s instructions. Gene-specific primers for qRT-PCR were designed with Primer 6.0 (Table 1). A single peak of the melting curve in qRT-PCR was used to ensure the specificity of the primers (Appendix A). The reaction mixture utilized Taq Pro Universal SYBR qPCR Master Mix (Vazyme Biotech, Nanjing, China) as per the manufacturer’s instructions. All qRT-PCR reactions were carried out using the QuantStudio™ Real-Time PCR system under the following conditions: 95 °C for 3 min, followed by 40 cycles of 95 °C for 15 s, 56 °C for 15 s, and 72 °C for 15 s. Melting curves were recorded after the 40th cycle by incrementally increasing the temperature by 0.5 °C every 5 s from 65 °C to 95 °C. The *EF-1α* gene was used as the internal control for normalization [65]. Relative gene expression levels were calculated using the 2^−ΔΔCT^ method [66].

### 4.8. Statistical Analysis

All data were first analyzed for normality of distribution using the Kolmogorov–Smirnov test of normality, and homogeneity of variance using the Levene homogeneity of variance test. The data are presented as the mean ± standard deviation (SD) from at least three independent replicates. Statistical analyses were conducted using Tukey’s HSD test at *p* ≤ 0.05, performed with the SPSS statistical package (version 17.0, Chicago, IL, USA). Charts were created using GraphPad Prism 8 (San Diego, CA, USA).

## 5. Conclusions

This study demonstrated that nitric oxide (NO) pretreatment effectively promoted the release of dormancy in lily bulblets by modulating the antioxidant system, endogenous hormone levels, and carbohydrate metabolism. The findings confirmed that SNP treatment enhanced the enzymatic activity of endogenous NOS-like enzymes and nitrate reductase (NR), thereby increasing the production of endogenous NO, while c-PTIO treatment had the opposite effect. The 10 mM SNP treatment reduced the dormancy time by 16%. The effect of 1 mM of c-PTIO treatment is consistent with providing an additional 20% cold storage time. This approach for pre-treating lily bulblets not only regulated the flowering period but also reduced the cold storage time required, which was particularly important for minimizing greenhouse gas emissions.

## Figures and Tables

**Figure 1 ijms-26-00156-f001:**
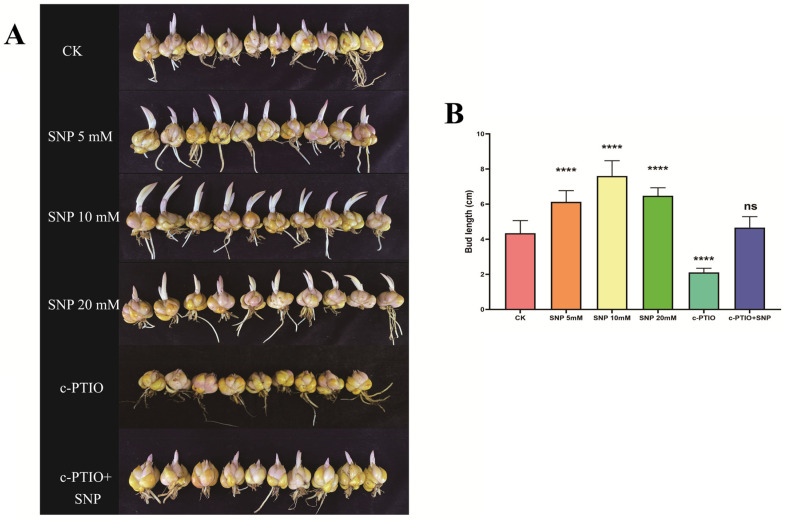
Lily bulblet dormancy release was affected by SNP and c-PTIO. (**A**) Phenotypes of bulblets treated with exogenous SNP (5 mM, 10 mM, and 20 mM) and c-PTIO (1 mM) for 3 h and grown at room temperature (4 °C) for 60 days. (**B**) Bulb length statistics. The data are shown as three independent biological replicates and three technical replicates (*n* = 3). ****—*p* ≤ 0.0001, ns—not significant.

**Figure 2 ijms-26-00156-f002:**
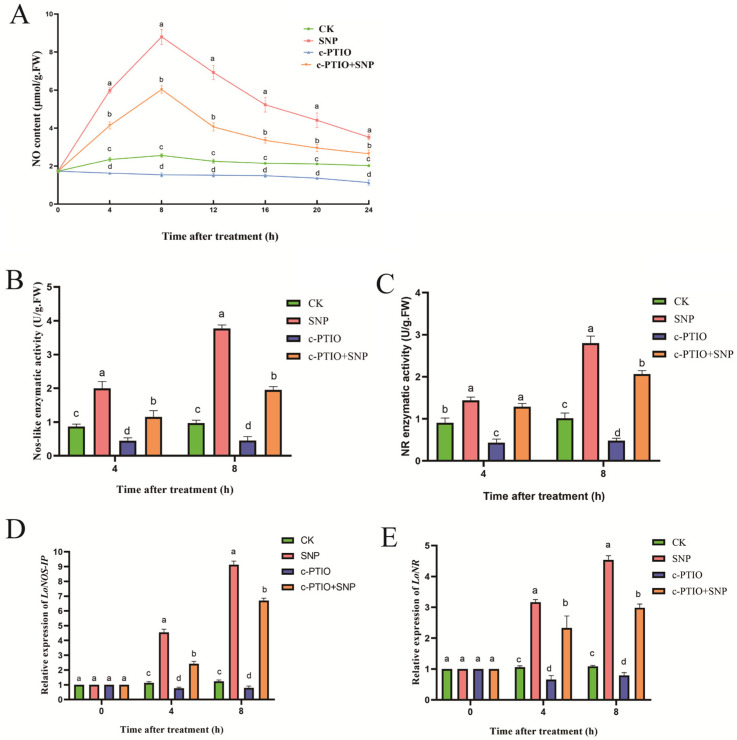
Effects of SNP and c-PTIO on NO contents (**A**), no enzyme activity (**B**,**C**), and NO-related genes expression (**D**,**E**). Values are means ± SDs (*n* = 3). The data are shown as three independent biological replicates and three technical replicates (*n* = 3). Different lowercase letters indicate statistically significant differences at *p* ≤ 0.05.

**Figure 3 ijms-26-00156-f003:**
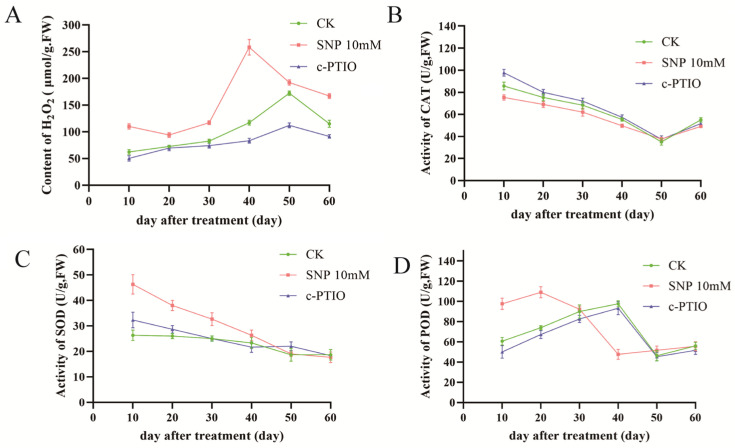
Changes in antioxidant enzyme activity (**B**–**D**) and H_2_O_2_ contents (**A**) of lily bubs treated with 10 mM SNP, 1 mM c-PTIO or SNP with c-PTIO during dormancy release at room temperature (4 °C) for 60 days. The data are shown as three independent biological replicates and three technical replicates (*n* = 3).

**Figure 4 ijms-26-00156-f004:**
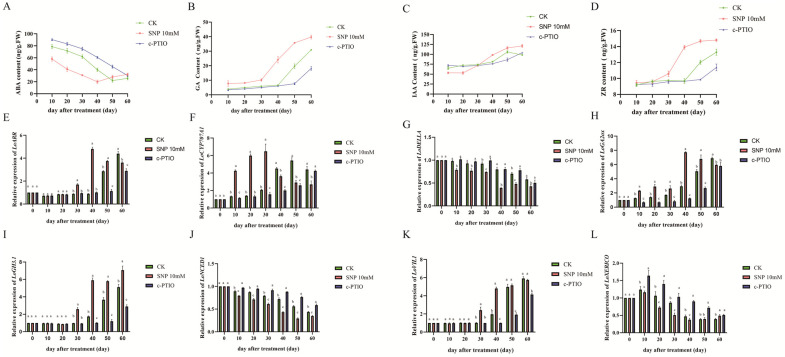
Changes in endogenous phytohormone contents and related genes expression of lily bubs treated with 10 mM SNP, 1 mM c-PTIO or SNP with c-PTIO during dormancy release at room temperature (4 °C) for 60 days. (**A**) ABA contents; (**B**) GA contents; (**C**) IAA contents; (**D**) ZR contents; (**E**) relative expression of *LoARR*; (**F**) relative expression of *LoCYP707A1*; (**G**) relative expression of *LoDELLA*; (**H**) relative expression of *LoGA20x*; (**I**) relative expression of *LoGH3.1*; (**J**) relative expression of *LoNCED1*; (**K**) relative expression of *LoVIL1*; (**L**) relative expression of *LoXERICO*. The data are shown as three independent biological replicates and three technical replicates (*n* = 3). Different lowercase letters indicate statistically significant differences at *p* ≤ 0.05.

**Figure 5 ijms-26-00156-f005:**
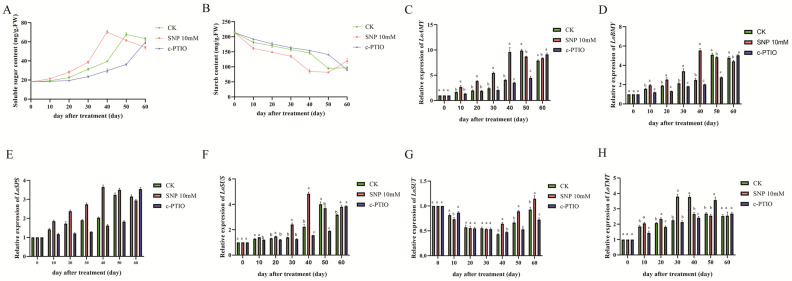
Changes in soluble sugar contents, starch contents, and related genes expression of lily bulbs treated with 10 mM SNP, 1 mM c-PTIO or SNP with c-PTIO during dormancy release at room temperature (4 °C) for 60 days. (**A**) Soluble sugar contents; (**B**) starch contents; (**C**) relative expression of *LoAMY*; (**D**) relative expression of *LoBMY*; (**E**) relative expression of *LoSPS*; (**F**) relative expression of *LoSUS*; (**G**) relative expression of *LoSUT*; (**H**) relative expression of *LoTMT*. The data are shown as three independent biological replicates and three technical replicates (*n* = 3). Different lowercase letters indicate statistically significant differences at *p* ≤ 0.05.

**Figure 6 ijms-26-00156-f006:**
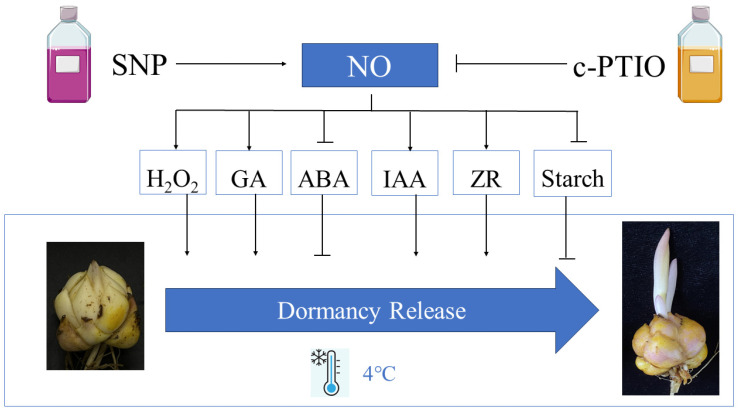
Model depicting how NO regulates bulblet dormancy release. Exogenous SNP treatment activates NOS-like or NR activity, which promotes NO production, while c-PTIO treatment has the opposite effect. NO promotes the contents of H_2_O_2_, GA, IAA, ZR for accelerating the progress of dormancy release. NO also promotes the catabolism of ABA and starch. The resulting decreased contents of ABA and starch ultimately lead to dormancy release.

**Table 1 ijms-26-00156-t001:** Genes and primer sets used for qRT-PCR analysis.

Gene Name	Protein Description	Primer Sequence	Length (bp)
*LoNOS-IP*	Nitric oxide synthase-interacting protein	F: CCCTCAACCCTTTCACAG R: CACTTTGTCCTTGTCCTTAC	101
*LoNR*	Nitrate reductase	F: TGTTCGTCTCGCAGTAAG R: TGTTCGTCTCGCAGTAAG	100
*LoARR*	Two-component response regulator ARR-A family	F: ATATGCTGATGCTGCTCTC R: CTCAACTCGGAATGGTGAT	145
*LoCYP707A1*	Cytochrome P450, family 707, subfamily A, polypeptide 1	F: GAAGAAGCAGAAGAAGTATGG R: GCACCTGAGACAAGAACA	108
*LoDELLA*	DELLA protein	F: GAAGCACTCCACTACTACTC R: CCTCGCATCCAATAACATTC	142
*LoGA20x*	Gibberellin 20 oxidase	F: TGGTTATCACGGTGTAGGA R: GGAAGCGAAGTTGGAGTT	139
*LoGH3.1*	Auxin-responsive GH3 family protein	F: ACACTGCTGCCGAATATG R: CACCTCTACCTCCACCAA	150
*LoNCED1*	9-cis-epoxycarotenoid dioxygenase	F: TTCGTTCATCGGAGATTGT R: CGGATTGTGTTAGGTTAGTG	118
*LoVIL1*	Vernalization insensitive like1	F: TCATCCTCAACCTCCTCTTA R: CAAGTTCAGGCAGTATTCG	130
*LoXERICO*	RING/U-box superfamily protein	F: GACAAGCGAGGTAGTGAG R: TTAGTTGAGAGCCGATCTG	112
*LoAMY*	Alpha-amylase	F: ATGGAATGGAAGTTCTCTGA R: CTGAAGTGTGGACTGGTT	122
*LoBMY*	Beta-amylase	F: ACGGTAAGCAGGTGATTG R: GATGACGCCAAGAGGAAG	101
*LoSPS*	Sucrose phosphate synthase	F: GGAGGACATCAATGCTACA R: CCACTGCTCTTCAATCTCT	115
*LoSUS*	Sucrose synthase	F: CAAGAAGGTCAAGGAGCAGATG R: CCGCACTAAGGAAAGCAGAG	159
*LoSUT*	Sucrose transport protein	F: TTATGGCTCTCTGCTTTGTA R: TGTGCGAGTAGAAATCATTG	182
*LoTMT*	Tonoplast Monosaccharide Transporter	F: TTGGCTCTGGATCGCTATCG R: CTCGCTCTCACTGTCACTCTC	94

F means forward primer; R means reverse primer.

## Data Availability

All data, tables, and figures in this manuscript are original.

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
