# Peer review of "Nitric Oxide Pre-Treatment Advances Bulblet Dormancy Release by Mediating Metabolic Changes in Lilium"

_ijms, 2024, doi:10.3390/ijms26010156_

Round 1
Reviewer 1 Report
Comments and Suggestions for Authors
The study “Nitric Oxide Pre-Treatment Advances Bulblet Dormancy Release by Mediating Metabolic Changes in Lilium” integrates physiological, biochemical, and molecular analyses, highlighting NO's role in modulating antioxidant activity, hormone levels, and carbohydrate metabolism. While the research addresses an important aspect of lily cultivation, there are areas where the methodology, data presentation, and discussion could be enhanced to improve clarity, depth, and scientific rigor. The following comments aim to provide constructive feedback to refine the manuscript for improved impact and readability.
- Include a brief mention of specific environmental and economic benefits in the introduction to emphasize the practical significance of reducing cold storage duration.
- Clarify why the specific concentrations of SNP and c-PTIO were chosen in the plant material and applied treatment section. Provide references if based on preliminary trials or literature.
- Specify whether bud length directly correlates with dormancy release across all treatments or if additional parameters, like sprouting rate, were also considered.
- Add context on the role of NR and NOS-like enzymes in NO production to justify enzyme activity measurements. - Explain the relevance of Hâ‚‚Oâ‚‚ content to dormancy release, particularly its role in oxidative stress management. - Link the selected genes for qRT-PCR analysis to dormancy and NO signaling based on previous studies or known pathways.
- Provide more quantitative details in the results section, such as fold changes in enzyme activities, NO content, or gene expression levels, for better clarity.
- Reiterate specific quantitative outcomes, such as the percentage reduction in dormancy period, in the conclusions to make them more impactful.
- Indicate whether data were tested for normality and homogeneity of variance before applying ANOVA in the statistical analysis section.
Comments on the Quality of English Languageplease check overall grammar consistency.
Author Response
The study “Nitric Oxide Pre-Treatment Advances Bulblet Dormancy Release by Mediating Metabolic Changes in Lilium” integrates physiological, biochemical, and molecular analyses, highlighting NO's role in modulating antioxidant activity, hormone levels, and carbohydrate metabolism. While the research addresses an important aspect of lily cultivation, there are areas where the methodology, data presentation, and discussion could be enhanced to improve clarity, depth, and scientific rigor. The following comments aim to provide constructive feedback to refine the manuscript for improved impact and readability.
Comment 1: Include a brief mention of specific environmental and economic benefits in the introduction to emphasize the practical significance of reducing cold storage duration.
Response: The practical significance of cold storage has been clearly added in the introduction section (lines 39-46, 114-117).
Comment 2: Clarify why the specific concentrations of SNP and c-PTIO were chosen in the plant material and applied treatment section. Provide references if based on preliminary trials or literature.
Response: The appropriate concentrations of SNP and c-PTIO were determined based on preliminary experiments, as detailed in Supplementary Table S1. When the concentration of NO was below 5 mM, its effect on dormancy release was comparable to that of the control bulbs. However, bulbs treated with NO concentrations exceeding 20 mM began to rot during storage. Similarly, c-PTIO treatments at concentrations below 1 mM had no significant effect on dormancy release, while concentrations above 1 mM inhibited it and led to bulb rotting during storage. Based on these observations, we selected suitable concentrations of SNP and c-PTIO for further experiments. Appropriate corrections have been made in lines 328–329.
Comment 3: Specify whether bud length directly correlates with dormancy release across all treatments or if additional parameters, like sprouting rate, were also considered.
Response: The bud length of lily bulbs has been widely used as a standard indicator for determining dormancy release timing, as referenced in previous studies [21,45] and described in lines 122–124. In our earlier work [29], we demonstrated that the dynamic transformation of callose—its degradation and deposition—controls the opening and closing of plasmodesmata (PD) channels, thereby regulating dormancy release and maintenance. Low-temperature exposure was shown to trigger the growth transition in lily bulbs, a process accompanied by PD opening, accelerated intercellular communication, and increased substance transport, ultimately leading to bud growth. Furthermore, bulbs sprout normally after accumulating sufficient cold exposure. Based on these findings, we conclude that bud length is the most reliable indicator for determining dormancy release timing.
Comment 4: Add context on the role of NR and NOS-like enzymes in NO production to justify enzyme activity measurements. - Explain the relevance of H2O2 content to dormancy release, particularly its role in oxidative stress management. - Link the selected genes for qRT-PCR analysis to dormancy and NO signaling based on previous studies or known pathways.
Response: The roles of NR and NOS-like enzymes in NO production have been included in lines 82–93 to support the justification for enzyme activity measurements. To clarify the relevance of Hâ‚‚Oâ‚‚ content to dormancy release, particularly its role in oxidative stress management, we have made revisions in lines 63–70, highlighting the effect of Hâ‚‚Oâ‚‚ on dormancy release based on previous studies. In the discussion section, we focused on the relationships among the selected genes, dormancy, and NO. In lines 270–293, we examined the role of plant hormone-related genes in dormancy, drawing on previous studies, and discussed how dynamic changes in endogenous NO influence these hormone-related genes. In lines 294–311, we explored the role of carbohydrate metabolism-related genes in dormancy, again referencing previous research, and further discussed how fluctuations in endogenous NO levels affect the expression of carbohydrate-related genes.
Comment 5: Provide more quantitative details in the results section, such as fold changes in enzyme activities, NO content, or gene expression levels, for better clarity.
Response: The results have been improvised following the suggestion.
Comment 6: Reiterate specific quantitative outcomes, such as the percentage reduction in dormancy period, in the conclusions to make them more impactful.
Response: We have made necessary corrections in lines 418- 420.
Comment 7: Indicate whether data were tested for normality and homogeneity of variance before applying ANOVA in the statistical analysis section.
Response: All data were first analyzed for normality of distribution using the Kolmogorov–Smirnov test of normality, and homogeneity of variance using the Levene homogeneity of variance test (see lines 406-408).
Reviewer 2 Report
Comments and Suggestions for Authors
The research was concentrated on the effects of NO on dormancy release in lily bulblets using SNP and c-PTIO. The most important results showed that SNP treatment promoted dormancy release, while c-PTIO inhibited it. Measurement of endogenous NO levels in the bulbs, along with enzyme activities of NOS-like and NR and gene expression levels of LoNOS-IP and LoNR, confirmed that NO plays a role in promoting dormancy release in lilies. The manuscript contains interesting research results; hence, it may be considered for further stages of evaluation and minor revisions.
In my opinion, the following revisions should be included:
- Regarding the gene expression studies, the Authors did not use the highly specific molecular fluorescent probes. It was used a SYBR Green dye; therefore, it should be added the graphical evidence that there were no additional (non-specific) PCR amplicons. I strongly recommend including the results of post-PCR analyses, i.e. melting curve analyses of the PCR products.
-Lines 302-303: The Authors should sufficiently explain in the manuscript, on what basis were selected SNP solutions at concentrations of 5 mM, 10 mM, and 20 mM, as well as a 1 mM solution of c-PTIO.
- There is no information how the amount and spectral purity of total RNA was assessed (RIN value? Capillary electrophoresis? Agarose gel electrophoresis? Spectrophotometric evaluation?)
- Line 338: The name and catalogue number of an enzyme-linked immunosorbent assay (ELISA) kit should be added.
- I suggest the Tukey’s test instead of the Duncan’s test. The name of the statistical test should be added in the caption of each figure.
Author Response
The research was concentrated on the effects of NO on dormancy release in lily bulblets using SNP and c-PTIO. The most important results showed that SNP treatment promoted dormancy release, while c-PTIO inhibited it. Measurement of endogenous NO levels in the bulbs, along with enzyme activities of NOS-like and NR and gene expression levels of LoNOS-IP and LoNR, confirmed that NO plays a role in promoting dormancy release in lilies. The manuscript contains interesting research results; hence, it may be considered for further stages of evaluation and minor revisions.
Comment 1: Regarding the gene expression studies, the Authors did not use the highly specific molecular fluorescent probes. It was used a SYBR Green dye; therefore, it should be added the graphical evidence that there were no additional (non-specific) PCR amplicons. I strongly recommend including the results of post-PCR analyses, i.e. melting curve analyses of the PCR products.
Response: Thank you for your suggestion. we have made appropriate corrections in line 392- 393, and provide graphical evidence in supplementary file Table S1.
Comment 2: Lines 302-303: The Authors should sufficiently explain in the manuscript, on what basis were selected SNP solutions at concentrations of 5 mM, 10 mM, and 20 mM, as well as a 1 mM solution of c-PTIO.
Response: The appropriate concentrations of SNP and c-PTIO were determined based on a preliminary experiment, with reference to the supplementary file Table S1. When the concentration of NO was below 5 mM, the dormancy release effect was consistent with that of the control bulbs. However, bulbs treated with NO concentrations above 20 mM showed signs of rotting during storage. The results also indicated that c-PTIO treatment at concentrations below 1 mM did not affect dormancy release, while concentrations above 1 mM inhibited dormancy release. Furthermore, bulbs treated with more than 1 mM of c-PTIO exhibited rotting during storage. Therefore, we selected the optimal concentrations of SNP and c-PTIO for the subsequent experiments. Necessary corrections have been made in lines 314–315.
Comment 3: There is no information how the amount and spectral purity of total RNA was assessed (RIN value? Capillary electrophoresis? Agarose gel electrophoresis? Spectrophotometric evaluation?)
Response: RNA degradation and contamination was monitored on 1% agarose gel. RNA quantification and purity assessment were performed spectroscopically using the NanoPhotometer® spectrophotometer (IMPLEN, CA, USA) (see lines 387-389).
Comment 4: Line 338: The name and catalogue number of an enzyme-linked immunosorbent assay (ELISA) kit should be added.
Response: The catalogue numbers of used kits have been added. Please see lines 367-368.
Comment 5: I suggest the Tukey’s test instead of the Duncan’s test. The name of the statistical test should be added in the caption of each figure.
Response: Thank you for your suggestion. We reassessed the statistical significance using ANOVA followed by Tukey’s HSD test and obtained the same results. Additionally, we have included the statistical test details in each figure.
Round 2
Reviewer 1 Report
Comments and Suggestions for Authors
Manuscript can be accepted.
Comments on the Quality of English LanguageSufficient to understand.